# The Extraction and High Antiproliferative Effect of Anthocyanin from Gardenblue Blueberry

**DOI:** 10.3390/molecules28062850

**Published:** 2023-03-22

**Authors:** Fengyi Zhao, Jialuan Wang, Weifan Wang, Lianfei Lyu, Wenlong Wu, Weilin Li

**Affiliations:** 1Fruit Research Center, Institute of Botany, Jiangsu Province and Chinese Academy of Sciences, Nanjing 210014, China; 2College of Chemical Engineering, Nanjing Forestry University, Nanjing 210037, China; weifan@njfu.edu.cn; 3Co-Innovation Center for the Sustainable Forestry in Southern China, College of Forestry, Nanjing Forestry University, Nanjing 210037, China

**Keywords:** blueberry, anthocyanins, functional components, antiproliferative effect, apoptosis

## Abstract

Blueberries are rich in flavonoids, anthocyanins, phenolic acids, and other bioactive substances. Anthocyanins are important functional components in blueberries. We collected 65 varieties of blueberries to investigate their nutritional and functional values. Among them, Gardenblue had the highest anthocyanin content, with 2.59 mg/g in fresh fruit. After ultrasound-assisted solvent extraction and macroporous resin absorption, the content was increased to 459.81 mg/g in the dried powder. Biological experiments showed that Gardenblue anthocyanins (L^1^) had antiproliferative effect on cervical cancer cells (Hela, 51.98 μg/mL), liver cancer cells (HepG2, 23.57 μg/mL), breast cancer cells (MCF-7, 113.39 μg/mL), and lung cancer cells (A549, 76.10 μg/mL), and no apparent toxic effects were indicated by methyl thiazolyl tetrazolium (MTT) assay, especially against HepG2 cells both in vitro and in vivo. After combining it with DDP (cisplatin) and DOX (doxorubicin), the antiproliferative effects were enhanced, especially when combined with DOX against HepG2 cells; the IC_50_ value was 0.02 μg/mL. This was further evidence that L^1^ could inhibit cell proliferation by inducing apoptosis. The detailed mechanism might be L^1^ interacting with DNA in an intercalation mode that changes or destroys DNA, causing apoptosis and inhibiting cell proliferation. The findings of this study suggest that L^1^ extract can be used as a functional agent against hepatoma carcinoma cells.

## 1. Introduction

Blueberry (*Vaccinium* L.) is a new type of berry fruit tree that has been cultivated on a large scale in China. It is one of the five healthy fruits recommended by the World Food and Agriculture Organization because of its pleasant color, sweet and sour taste, and excellent nutrition including organic acids, phenolic acids, and flavonoids [1,2,3]. Blueberries are rich in bioactive compounds, mainly flavones and other polyphenolic compounds [4]. They have a wide range of pharmacological effects, including anticancer effects [5], antioxidant effects [6,7], antiviral effects [8], anti-inflammatory effects [9,10], hypoglycemic effects [11,12], improvement of dyslipidemia [13], anti-obesity effects [14,15] and the prevention and treatment of age-related degenerative diseases [16] and cardiovascular diseases [17]. Anthocyanins account for 50~80% of the total polyphenol content in blueberry fruits. Our group analyzed the anthocyanins in the Gardenblue blueberry type. We found that the main anthocyanin components were cyanidin and pelargonidin, which were glycosylated or acylated forms of glucose, galactose, and arabinose. Most studies have focused on the crude extract of blueberries, and it is still necessary to explore the exact effective substances with significant biological activity. The mechanism of their curative effect is also worth further study.

Anthocyanins are secondary metabolites generated via the synthesis pathways of phenylpropionic acid and flavonoids in plants [18]. Their basic structure consists of two benzene rings (C6-C3-C6) connected by a central three-carbon chain. The major common anthocyanins in plants include pelargonidin, cyanidin, delphinidin, peonidin, petunidin, and malvidin [19]. Usually, anthocyanins are formed through glycosidic bonds with one or more molecules of glucose, rhamnose, galactose, xylose, or arabinose, etc. The glycosides and hydroxyl groups can also form acylated anthocyanins through ester bonds with one or more molecules of coumaric acid, ferulic acid, caffeic acid, p-hydroxybenzoic acid, or other aromatic acids and fatty acids in anthocyanins. More than 500 anthocyanins are known in nature. Pharmacological studies suggest that anthocyanins possess physiological activities, such as promoting retinoid resynthesis, improving microcirculation, antiproliferative and anti-inflammation effects, and reducing blood glucose and blood lipid levels [20].

Cancer is a serious threat to human life and health. At present, the main tumor treatments are surgery, radiotherapy, and chemotherapy, but these are traumatic to the human body and have serious side effects. As a natural plant pigment, anthocyanins have biological activities, and the study of their antiproliferative effects has attracted increasing attention from researchers [21,22,23,24,25]. These effects are related to their excellent antioxidant activity, anti-DNA breakage effect, inhibition of cell proliferation, induction of apoptosis, anti-angiogenesis effect, and other activities. In many reports, anthocyanins have been considered to have effects against colon cancer, skin cancer, prostate cancer, leukemia, and breast cancer, among others. In recent years, many research results have shown that blueberry anthocyanin can inhibit the proliferation of cancer cells. Wang and co-workers found it had a significant inhibitory effect on mice melanoma cells (B16F10) in a dose-dependent manner in vitro [26]. Zhou et al. reported blueberry anthocyanin had a certain inhibitory effect on HepG2 cells and induced HepG2 cells apoptosis in a dose-dependent manner in vitro [27]. The mechanism of apoptosis is related to the mitochondrial apoptosis pathway, mitogen-activated protein kinase (MAPK) signaling pathway, p53 signaling pathway, apoptosis signaling pathway in the endoplasmic reticulum (ER) stress response, the TGF-β signaling pathway, and so on.

Due to the damage drugs cause cells, there are also studies on the inhibition effect of blueberry anthocyanin combined with current drugs on cancer cells and the protective effect on normal cells. One study showed that acylated blueberry anthocyanin combined with cyclophosphamide could increase the enzyme activities of total superoxide dismutase (T-SOD), cat, and glutathione peroxidase (GSH PX) in mouse cancer liver tissue [28]. Another study showed for the first time that blueberry anthocyanin has an inhibitory effect on human colon cancer cells (HCT-116), which is mainly related to the induction of apoptosis, cell cycle arrest of G0/G1, reactive oxygen species regulation, and reduction of matrix metalloproteinases [29]. Among these combined conventional drugs, doxorubicin (DOX, also termed Adriamycin) and cisplatin (DDP) have attracted more attention. DOX is widely used as a first-choice anticancer drug for many cancers and is one of the most effective anticancer drugs developed because of its apoptosis-inducing activity [30]. Similar to DOX, DDP exerts anticancer activity via multiple mechanisms, but its most acceptable mechanism involves generation of DNA lesions by interacting with purine bases on DNA followed by activation of several signal transduction pathways, which finally lead to apoptosis [31]. We chose drugs that combine DOX and DDP to enhance the antiproliferative effect of blueberry anthocyanin extract and to explore its possible mechanism related to apoptosis and DNA.

In this work, 65 varieties of blueberries were collected and their main phytochemicals and polyphenol qualities were investigated. We found that the content of Gardenblue anthocyanins was the highest. After further purification, we explored their inhibition of cancer cell proliferation both in vitro and in vivo, as well as the effect of combined drugs and the possible mechanism (Figure 1). We hoped to establish a theoretical basis for developing blueberries’ value and contribute to exploring future antiproliferative drugs from natural foods or plants through an easier extraction method. 

## 2. Results

### 2.1. Main Phytochemicals Analysis

A total of 65 varieties (lines) of blueberries were investigated for their phytochemical profiles, including soluble solids, organic acids, solidity–acid ratio, and total anthocyanin content (Table 1). 

A high content of soluble solids indicated that sugar content was high, which generally means the fruit is sweet. From Table 1, the average soluble solid content was 11.33%, and the content of Vernon was the highest with 15.63%, followed by Homebell (14.77%), Tifblue (13.93%), Beckyblue (13.73%), Powderblue (13.67%), Brightwell (13.6%), Pink Lemonade (13.57%), Anna (13.57%), Onslow (13.17%), Chandler (13.03%), and others. The high total acid content suggested the fruit is generally acidic. Caroline blue had the highest total acid content with 1.44% among the investigated blueberry varieties, followed by Sweetheart (1.0%), Ozarkblue (0.94%), Summit (0.87%), Biloxi (0.86%), Berkeley (0.80%), Sharpblue (0.79%), Baldwin (0.78%), Zhongzhi ‘2’ (0.77%), Springhigh (0.77%), and others. The solidity–acid ratio is the ratio of soluble solid to total acid content, higher content indicating sweeter fruit. The results showed Anna was the sweetest, with the highest solidity–acid ratio of 32.38, followed by Jewel (27.76), Beckyblue (27.74), Zhaixuan ‘4’ (27.11), Ruble (26.21), and others. Blueberries have abundant anthocyanins, which are the main component relevant to health. Anthocyanin content is an important index of blueberry fruit quality. According to the investigation of total anthocyanins, Gardenblue had the highest anthocyanin content with 259.46 mg/100 g. The other varieties of blueberries in the top five were Zhaixuan ‘4’ (229.89 mg/100 g), Vernon (228.93 mg/100 g), Centurion (220.85 mg/100 g), and Ruble (192.70 mg/100 g). 

We further investigated the main polyphenols of the Gardenblue blueberries, including phenols, anthocyanins, ellagic acid, and flavonoids (Table 2). The contents of Gardenblue’s main nutrients were all enhanced by nearly 60% by converting the fresh fruit into dried powder (extract, 459.81 mg/g). Anthocyanins represented the highest content of the main nutrients and are an important functional component in Gardenblue blueberries, belonging to the group of phenols. Lanmei 1 blueberries have the highest anthocyanin content on the domestic market, at 443.08 mg/g [32]. Gardenblue has a higher anthocyanin content (459.81 mg/g, dried powder) than Lanmei 1 and has more potential to be developed as a functional food product with added value.

### 2.2. Component Analysis

The analysis via UPLC-QTOF-MS^2^ showed that there were five types of anthocyanidin: delphintin (*m*/*z* 303), cyanidin (*m*/*z* 287.1), petunidin (*m*/*z* 317), malvidin (*m*/*z* 331.1), and peonidin (*m*/*z* 301.1), and the sugar residues associated with them were glucoside, galactoside, and arabinoside in Figure 1. Peak 5 and peak 7 had the same molecular ion peak (*m*/*z* 449.1) and molecular fragment (*m*/*z* 287.1). According to the order of peak production, it was inferred from the literature [33] that peak 5 was cyanidin-3-galactoside and peak 7 was cyanidin-3-glucoside. Peak 8 and peak 9 had the same molecular ion peak (*m*/*z* 479.1) and molecular fragment (*m*/*z* 317.1). According to the peak sequence and the literature [18], it was speculated that peak 8 was petunidin-3-galactoside, peak 9 was petunidin-3-glucoside, and peak 6 was delphintin-arabinoside or xyloside. Peak 14 and 16 had the same molecular ion peak (*m*/*z* 493.1) and molecular fragment (*m*/*z* 331), which were speculated to be malvidin-3-galactoside, malvidin-3-glucoside, and malvidin-3-arabinoside or xyloside, respectively. Peak 17 and peak 19 had the same molecular ion peak (*m*/*z* 433.1) and molecular fragment (*m*/*z* 303.1), which were formed by *m*/*z* 433.1 losing a neutral fragment with a mass number of 132. The latter has the same mass number as five-carbon sugars (arabinoside or xyloside) and peonidin after losing a water molecule. Other results are displayed in Table 3. In summary, the highest content of anthocyanins in blueberry fruit was malvidin-3-glucoside.

### 2.3. Biological Evaluation

#### 2.3.1. Antiproliferative Activities

In order to compare the antiproliferative activity of purified Gardenblue anthocyanin (L^1^) and well-known antiproliferative drugs, the effects of L^1^, DDP, DOX, L^1^-DDP (L^1^ combined with cisplatin), and L^1^-DOX (L^1^ combined with doxorubicin) on the growth of human cancer cell lines Hela, HepG2, MCF-7, and A549, and the normal cell HUVEC, were evaluated in vitro according to IC_50_ values. DDP and DOX were used as the positive control. The results are listed in Table 4.

According to the results, the toxicity of L^1^ was lower than DDP, DOX, L^1^-DDP, and L^1^-DOX for HUVEC cells. For Hela cells, the combined administration of L^1^ and DDP or DOX led to more effective anti-Hela activity than individual incubation. For the HepG2 cells, the IC_50_ value was smaller in total than the other four cells, especially L^1^ combined with DOX, which had the highest anti-HepG2 activity (0.02 μg/mL). L^1^ combined with DDP had higher activity against HepG2 cells than L^1^. For MCF-7 cells, the IC_50_ values of L^1^ combined with DDP (30.45 μg/mL) and DOX (6.97 μg/mL) were smaller than L^1^ (113.39 μg/mL) and DDP (39.88 μg/mL), which suggested L^1^ combined with both DOX and DDP had higher activity against MCF-7 cells than L^1^. For A549 cells, the IC_50_ values of L^1^ combined with DDP (47.07 μg/mL) and DOX (0.55 μg/mL) were smaller than L^1^ (76.10 μg/mL), which indicated that the combination of DDP or DOX had additive and synergistic effects on A549 cells. According to Wang’s report, blueberry malvidin-3-galactoside suppressed the development of hepatocellular carcinoma cells (HepG2) [34].

The cytotoxicity for the nonmalignant cell line (HUVEC cells) was also investigated to characterize the selectivity expressed according to selectivity index (SI) (SI = (IC_50_ for nonmalignant cell line HUVEC)/(IC_50_ for human tumor cell line)) [35], illustrated in Figure 2. SI was an important aspect for prospective pharmacological applications. DDP and DOX displayed good cytotoxicity to human tumor lines but relatively low selectivity (SI < 1). It was one of the important aims of our work to reduce cytotoxicity to nonmalignant cells. After being combined with L^1^, it was found that L^1^-DDP and L^1^-DOX demonstrated a moderate-to-good cytotoxic activity against human cancer cells and obviously enhanced selectivity towards HepG2, MCF-7, and A549 compared to DDP and DOX.

In general, whether L^1^ was combined with DOX or DDP, the antiproliferative activity was significantly enhanced compared to L^1^, which revealed that a combination of drugs can improve the treatment effect. Among all tested materials, L^1^ combined with DOX had the smallest IC_50_ value of 0.02 μg/mL against HepG2 cells, which suggested that L^1^ combined with DOX was more effective at inhibiting cultured HepG2 cell survival, and it may be a potential antiproliferative drug.

#### 2.3.2. Induction of Apoptosis

Blueberry anthocyanins have been found to play a role in mitochondrial-mediated apoptosis [34]. Apoptosis plays a core role in cancer because its induction in cancer cells is the key to successful treatment [35]. Apoptosis is controlled by genes of orderly cell death. It is not a passive process, but an active process involving a series of activation, expression, and regulation of genes. It is not a pathological condition or a phenomenon of self-injury, but is required to adapt better to the environment and is an essential death process [36]. The apoptosis assay can provide important information for studying the action mode of cells. In our previous work, L^1^ was found to have higher anti-HepG2 activity; therefore, we further studied whether L^1^ induced apoptosis in HepG2 cells. The flow cytometric analysis is shown in Figure 3.

After L^1^ was incubated with HepG2 cells with the concentration range from 12.5 to 50 μg/mL, the number of live cells reduced and that of apoptotic cells increased, exhibiting a dosage-dependent effect, as in Figure 3. When the concentrations of L^1^ were 12.5, 25, and 50 μg/mL, the early apoptotic rates were 21.1%, 40.8%, and 42.5%, and the late apoptotic rates were 13.7%, 20.9%, and 38.2%, respectively. During the induction of the apoptosis process, live cells tended to develop toward apoptotic cells with increased concentration. The reported results distinctly illustrated that L^1^ can inhibit cell proliferation by inducing apoptosis.

#### 2.3.3. Detection of Intracellular Reactive Oxygen Species (ROS)

As previously reported, excess cellular levels of ROS cause damage to proteins, nucleic acids, lipids, membranes, and organelles, which can lead to activation of cell death processes such as apoptosis [37]. There is powerful evidence that ROS are the underlying cause of many chronic diseases that involve enhanced intracellular oxidative stress [38]. The effect of representative L^1^ on cellular ROS levels in HepG2 cancer cells was detected using flow cytometry. HepG2 cells were incubated with L^1^ (25 μg/mL) for 48 h, and the results are shown in Figure 4. The percentage of ROS was 21.2%, while the control was 4.1%. After adding NAC (N-acetylcysteine), a known ROS inhibitor, the percentage fell back to 9.9%. Generally, these data indicated that L^1^ could induce the enhancement of ROS production, which might cause apoptosis.

#### 2.3.4. Confocal Fluorescence Images

The induced apoptosis process was made visible via confocal fluorescence images (Figure 5), since L^1^ was able to inhibit cell proliferation by inducing apoptosis.

The nucleus stained with PI (propidium iodide) presented red fluorescence while the cytoplasm stained with Annexin V-FITC presented green fluorescence. Normal cells were not stained by fluorescence, while apoptotic cells were stained only by green fluorescence, and necrotic cells were stained by both green and red fluorescence. In the negative control, there was scarcely any green and red fluorescence signal, which suggested the cells were in a normal state. After incubation with L^1^ (25 μg/mL) at 37 °C for 48 h, the green fluorescence signal was strong inside the cytoplasm and the red fluorescence signal was very weak, demonstrating the apoptotic ability of L^1^. 

#### 2.3.5. In Vivo Experiment

L^1^ was administered to mice in in vivo experiments for further investigation. In this experiment, tumor model mice with HepG2 cells injected intravenously with L^1^ (10 mg/kg) from day 1 to day 25, every 3 days, were used. 

As shown in Figure 6a–c, the volume and weight of tumor model mice were reduced clearly after injection with L^1^ (10 mg/kg) compared to PBS control. For L^1^, the average volume of the HepG2 tumor was 0.495 cm^3^, while the PBS control was 1.478 cm^3^. The weight of the tumor was 0.60 g when the PBS control was 1.69 g. The relative tumor proliferation rate (T/C) was 35.56% and the tumor inhibition rate was up to 64.86%. In Figure 6d, no significant toxicity was observed in the heart, liver, spleen, lung, kidney, or brain tissues of the mice injected with L^1^, which displayed no obvious changes in the morphology of these organs. In Figure 6e, compared to PBS control, CD31 immunohistochemical staining with tumor model mice was conducted on the 25th day after injecting L^1^ (10 mg/kg), and they exhibited an obviously decreased tumor angiogenesis rate, which showed L^1^ could inhibit tumor growth. As Wang reported, higher doses of blueberry anthocyanin had significant inhibitory effects on the migration and invasion activity of HepG2 cells at 80 mg/kg in vivo, accompanied by a decrease in cytoplasm and a sparse cell distribution and suppressed tumor growth [34]. Overall, these results clearly suggested that L^1^ had high anti-HepG2 activity both in vitro and in vivo experiments and demonstrated that L^1^ possesses great promise as a nontoxic antiproliferative drug and future clinical treatment option.

#### 2.3.6. DNA Binding Modes

Rowe et al. reported that when DNA damage is introduced into cells from exogenous or endogenous sources, there is an increase in the amount of intracellular ROS that is not directly related to cell death but related to apoptosis [39]. Intercalation is well known to strongly influence the properties of DNA and has been reported as a preliminary step in mutagenesis. Many drugs can induce apoptosis through intercalating or binding with DNA, such as DOX [40]. We further investigated whether L^1^ had similar activity to DOX in terms of DNA-binding properties. The Salmon Sperm DNA (fDNA) binding modes were evaluated with ethidium bromide (EB) fluorescence displacement experiments. EB had no perceptible emissions in buffer solution. After adding DNA, the fluorescence intensity improved, which was considered to be due to its strong intercalation with DNA base pairs. The intercalation of the compound with the base pairs of DNA was confirmed when the fDNA-EB emissions were decreased or quenched upon adding a compound [41,42,43]. As expected, the emission intensity reduced (shown in Figure 7) by adding L^1^ to fDNA-EB, which showed that L^1^ can bind to DNA at the sites occupied by EB and that it can interact with DNA via intercalation.

The interaction between L^1^ and DNA was studied using fluorescence spectroscopy. The results of the fluorescence spectrum showed that L^1^ could induce fluorescence quenching in the fDNA-EB system, and the degree of fluorescence quenching increased with the increase in concentration. The above tests indicated that L^1^ interacted with DNA in an intercalation mode. Compounds can interact with DNA in an intercalation mode, which can change or destroy DNA, thus causing apoptosis, which is also one of the reasons for the inhibition of cell proliferation activity of L^1^.

## 3. Discussion

The solidity–acid ratio is one of the important indexes of fruit quality, which largely depends on the kinds of sugar contained in the fruit class, quantity, and organic acid content. High-acid and low-sugar fruit is experienced as sour and is poorly accepted by people, and low-acid and high-sugar fruit tastes weak and does not meet the requirements for fresh food. In addition, fruit sweetness is related to the type of sugar. Consumers may choose fresh blueberries based on their appearance, sweetness and acidity, taste, flavor, and overall feeling. The Gardenblue blueberry has become a popular choice among consumers and has several advantages, including a higher yield, medium fruit size, longer maturation period, and better taste, smell, and storage qualities. The most important thing to consider is that the Gardenblue blueberry has the highest anthocyanin content with 2.59 mg/g of fresh fruit and 459.81 mg/g of dried powder compared to other blueberries collected in the same period and place. Anthocyanin can inhibit the proliferation of cancer cells, which suggests Gardenblue anthocyanin has higher antiproliferative effects.

The antiproliferative effect of Gardenblue anthocyanin (L^1^) on cancer cells has been evaluated. Compared to three other cancer cells, L^1^ has the highest antiproliferative effect on HepG2 cells. We further proved that a combination of drugs can improve the treatment effect by using L^1^ combined with DOX and obtaining the smallest IC_50_ value of 0.02 μg/mL against HepG2 cells. According to the analysis of the selectivity index, it was found that L^1^-DDP and L^1^-DOX demonstrated a moderate-to-good cytotoxic activity against human cancer cells and obviously enhanced selectivity towards HepG2, MCF-7, and A549 cells compared to DDP and DOX, especially towards HepG2 cells. This result suggested that L^1^ combined with DOX was more effective at inhibiting cultured HepG2 cell survival, and it may be a potential antiproliferative drug. In vivo experiments indicated that anthocyanins can prevent and treat cancer, with a relative tumor proliferation rate (T/C) of 35.56% and a tumor inhibition rate of up to 64.86%, which is higher than the result in our previous paper covering the anticancer activity of natural products, suggesting that it has excellent antiproliferative effects [30,36]. L^1^ could induce the enhancement of ROS production, which might cause apoptosis. The further mechanism might be that L^1^ interacted with DNA in an intercalation mode, which can change or destroy DNA, thus causing apoptosis and inhibiting cancer cell proliferation. As Wang reports, blueberry anthocyanin suppressed the proliferation, polarization, migration, and invasion activities of HepG2 cells in vitro by regulating the protein expression of cyclin D1/B/E, caspase-3, cleaved caspase-3, Bax, c-Jun N-terminal kinase (JNK), and p-p38, activating phosphatase and tensin homologue deleted on chromosome 10 (PTEN), accompanied by a decrease in the phosphorylation-AKT (p-AKT) level, and lowering the protein expression levels of matrix metalloproteinase 2 (MMP-2) and matrix metalloproteinase 9 (MMP-9). In vivo, blueberry anthocyanin promoted the apoptosis of liver tumor cells, as determined by immunohistochemistry [34]. Similarly, Yang et al. found that blueberries had a very significant inhibitory effect on breast, non-small-cell lung, and colon cancer cells, and explored the possible mechanisms for this, such as apoptosis induction and inhibiting cell proliferation. A combination of suboptimal concentrations of equimolar anthocyanidins of berries suppressed the growth of two aggressive non-small-cell lung cancer cell lines, and a variety of berry mixtures with diverse anthocyanins had a therapeutic effect in non-small-cell lung cancer and prevented its future recurrence and metastasis. For cancer prevention, blueberries’ ability to decrease DNA damage (the first step of cancer) is seen as promising. Bioactive compounds of blueberries can regulate the expression of genes following DNA damage [5]. Blueberry anthocyanins may be recommended for the treatment of HepG2 cells. Besides anthocyanins, ellagitannins, and their gut microbiota-derived metabolites, other important bioactive molecules found in blueberries were shown to trigger autophagy in human colorectal cancer cells and to induce apoptosis by increasing the expression of proapoptotic proteins p21 and p53 and decreasing the anti-apoptotic protein expression of B-cell lymphoma-2 (Bcl-2). Moreover, the proapoptotic effect was achieved through downregulation of the phosphatidylinositol 3-kinase/amino threonine protein kinase (PI3K/AKT) signaling pathway [44].

With the rise in consumer awareness regarding the need for healthy eating habits, researchers have been striving to find alternative natural sources of additives that, while being completely safe, may also have health benefits. Gardenblue contains five anthocyanins (delphinidin, peonidin, petunidin, cyanidin, and malvidin) and can inhibit the growth of cancer cells and induced apoptosis, which suggests that it may be a potential healthy food or drug. 

## 4. Materials and Methods

### 4.1. General Materials

A total of 65 varieties of blueberries were collected during the ripening period (from May to July) in 2022 from the Institute of Botany, Jiangsu Province, and the Chinese Academy of Sciences, Baima field. Analytical reagents (AR) and solvents were used in the general experimental procedure unless otherwise stated. Purifications were performed using flash chromatography on microporous resin (HPD-100B). UV–vis adsorption spectra were recorded by a TU-1810 spectrophotometer. Annexin V-FITC/PI was purchased from Shanghai Beyotime Biotechnology Co. LTD, Shanghai, China. Reagents and compounds, including MTT, DMEM, FBS, and penicillin/streptomycin, were all commercially purchased from Nanjing Keybionet Biotechnology Co. LTD. Hela, HepG2, MCF-7, A549, and HUVEC cells were purchased from the National Collection of Authenticated Cell Cultures.

Other equipment included an EX-200A Electronic Balance, ZD-2 Automatic Potential Titrator, PL-5-B Low-speed Centrifuge, Philips Blender HR2838, Agilent 1260UHPLC-6530 Q-TOF MS, NIB610 microscope, KQ-300DE ultrasonic cleaner, BD Accuri C6 flow cytometry, and Zeiss LSM 900 Laser Scanning Confocal Microscope. 

### 4.2. Soluble Solids Content

Pal-1 was used to measure the soluble solid content of 65 varieties of fresh blueberry fruit (unit: °Brix).

### 4.3. Titratable Acid Content Determination

For the Determination of Total Acid in Food method from GB/T12456-2008, the determination result was calculated using citric acid. A total of 50 g of fresh blueberries was broken up and 3 g was taken out and placed into a 50 mL centrifuge tube whereupon 20 mL of deionized water was added. The mixture underwent ultrasonic treatment at 35 °C and 60 Hz for 20 min and was centrifuged at 5000 rpm for 5 min. The supernatant was transferred to a 100 mL beaker and the initial pH value of the solution was measured using a pH meter. The NaOH standard solution (0.1 mol/L) was titrated until the pH value of the solution was 8.0, which was the end point. The amount of NaOH was recorded and the titratable acid content was calculated [36].

### 4.4. Extraction and Purification

Improving upon previous work, 2 kg of frozen Gardenblue blueberry fruits were hydrolyzed with 0.02% pectinase at 40–50 °C for 2 h and then centrifuged at 4000 rpm for 5 min. We extracted the precipitates with 50% ethanol (containing 0.1% formic acid by volume) three times. After ultrasound-assisted extraction at 40 °C and 60 Hz for 30 min, the extract was centrifuged at 5000 rpm for 5 min [20,36].

We filtrated the above extract, retaining the supernatant; evaporated the ethanol, retaining the aqueous solution; and adsorbed with a macroporous resin column (HPD-100B). The solvent was eluted with 50% ethanol. We repeated the adsorption, collected the eluent, removed the solvent, and freeze dried the resulting substance. The final product obtained was Gardenblue anthocyanins (L^1^). 

### 4.5. Component Analysis 

An Agilent 1260 Infinity ultra-high-performance liquid chromatography system was used for separation according to Agilent Poroshell 120 SB-AQ (3.0 × 100 mm, 2.7 μm) columns. The mobile phase consisted of 0.1% formic acid water (A) and acetonitrile (B), using gradient elution: 0 min, 10%B; 15 min, 15%B; 20 min, 15%B; 30 min, 30%B; 35 min, 40%B; 36 min, 90%B; and 41 min, 90%B. The flow rate was 0.5 mL/min and the column temperature was 30 °C. The detector was an Agilent 6530 Accurate-Mass Tandem Quadrupole-Time of Flight Mass Spectrometer (Q-TOFMS) with an electrospray ion source (ESI). The positive ion mode was used to detect the mass spectrometry. The detection parameters were DAD wavelength: 280 nm, capillary voltage: +4.0 kV, sprayer pressure: 50 psi, dry temperature: 350 °C, Vcap: 3500 V, breakage voltage: 150 V, and scanning range: *m/z* 100–1000, and Nitrogen was used as the atomizing and desolvating gas with a flow rate of 1 L/min. The reference standards used to identify the compounds were cyanidin-3-O-glucoside, delphintin-3-O-glucoside, malvidin-3-glucoside, peonidin-3-glucoside, and petunidin-3-glucoside. 

### 4.6. Cell Culture, Antitumor Activities, and Cytotoxicity Assay

Hela, HepG2, MCF-7, A549, and HUVEC cells were seeded in 96-well plates with a density of 1 × 10^4^ cells per well. After 12 h of incubation at 5% CO_2_ and 37 °C, the culture media were removed and the cells were incubated with DDP, DOX, L^1^-DDP (L^1^ combined with cisplatin), and L^1^-DOX (L^1^ combined with doxorubicin) dissolved in DMEM at different concentrations (each concentration was repeated three times) for 48 h at 5% CO_2_ and 37 °C. Subsequently, we removed the culture media, and the new culture medium containing MTT (1 mg/mL) was added, followed by incubation for 4 h to allow the formation of formazan dye. After removing the medium, 200 μL DMSO was added to each well to dissolve the formazan crystals. Absorbance was measured at 595 nm in a microplate photometer. Cell viability values were determined (at least three times) according to the following formula: cell viability (%) = the absorbance of experimental group/the absorbance of blank control group × 100% [30,35,36].

### 4.7. Induction of Apoptosis Assay

We further investigated whether L^1^ could induce apoptosis. DMSO was used as a negative control. HepG2 cells (1 × 10^6^) were cultured in 35 mm dishes and incubated at 37 °C for 24 h. After incubation with DMSO at 5 μg/mL, L^1^ at 12.5, 25, and 50 μg/mL was added for 48 h (each concentration was repeated three times and the incubation time was the optimum), and then the treated cells were washed, trypsinized (non-EDTA, ethylene diamine tetraacetic acid), and centrifuged (2000 rpm/min). Next, the cells were collected and resuspended in 500 μL buffer solution loaded with Annexin V-FITC apoptosis detection reagent (with 5 μL Annexin V-FITC and 5 μL PI). The Annexin V-FITC stained cells were incubated for 5–15 min in the dark, and approximately 1 × 10^4^ cells were collected and 80,000 events for flow cytometry analysis with a single 488 nm argon laser were conducted with BD Accuri C6 flow software and cytometry [36].

### 4.8. Detection of Intracellular Reactive Oxygen Species (ROS)

HepG2 cells were cultured in a 6-well plate overnight and then incubated with L^1^ (0, 25 μg/mL) for 48 h. The cells were then incubated with dihydrodichlorofluorescein diacetate (DCFH-DA) (10 μM) for 30 min at 37 °C in the dark. Finally, the cells were collected and washed in PBS and the samples were analyzed via flow cytometry. Excitation wavelengths were 488 nm and emission wavelengths were 535 nm [36]. 

### 4.9. Confocal Fluorescence Images

HepG2 cells were seeded on 35 mm glass dishes for 24 h and then incubated with L^1^ (25 μg/mL) for 48 h at 5% CO_2_ and 37 °C before undergoing fluorescence imaging. After incubation, the culture medium was removed and cells were washed with PBS (pH = 7.4) and then incubated with fresh medium. Confocal fluorescence images were taken with the excitation of Annexin V and PI channels [36].

### 4.10. In Vivo Experiment

The in vivo experiment was undertaken using Nanjing Keygen Biotech. Co. Ltd., Nanjing, China. To develop the tumor model, 1 × 10^6^ HepG2 cells were subcutaneously injected into the right armpit of Balb/C nude mice. Two groups of HepG2-tumor-bearing mice with five mice per group were randomly chosen in our experiment: (1) PBS (as a control), (2) L^1^. After the size of the tumors reached 80 mm^3^, all agents, including PBS and L^1^ solutions, were administrated via an intravenous injection (dose = 10 mg/kg). During the next 25 days, the tumor size of each mouse in our experiments was measured with a vernier caliper every 3 days. To accurately evaluate the growth inhibition of tumors, the mice were sacrificed after 25 days, and then their tumors were collected, photographed, and weighed. Sections of tumor, heart, kidney, liver, lung, and spleen tissues of different groups harvested on the 25th day were observed using H&E staining and then examined by a pathologist. The tumor size was calculated as the volume = 0.5 × (tumor length) × (tumor width)^2^. The inhibition efficiency of tumor growth was calculated according to the equation: inhibition efficiency (%) = (1 − the weight of experimental group/the weight of control group) × 100%. CD31 immunohistochemical staining with mice was conducted on the 25th day after an intravenous injection of L^1^ (10 mg/kg) and PBS control [36].

### 4.11. DNA Binding Modes

A volume of 3 mL of DNA-EB mixture solution (C_DNA_/C_EB_ = 10) was added to the sample pool, and an equal volume of compound sample was added to the sample pool each time, which caused an increase in the concentration ratio between L^1^ and DNA. Fluorescence spectra were measured at an excitation wavelength of 520 nm [36].

### 4.12. Data Analysis

Statistical software SPSS 16.0 (IBM, Chicago, IL, USA) was used to analyze the data, including the correlation analysis. One-way ANOVA was used to compare mean values, and Duncan post hoc multiple comparisons were used (*p* < 0.05). Origin 2019b (OriginLab Crop., Northampton, MA, USA) was used to analyze the IC_50_ values with linear fitting. All experiments were repeated three times, and the results represent the mean ± standard deviation.

## 5. Conclusions

In this study, Gardenblue blueberries were found to have rich phytochemicals, such as polyphenols, anthocyanin, ellagic acid, and flavonoid. The highest content of anthocyanins in blueberries was malvidin-3-glucoside. Gardenblue anthocyanin extract had high antiproliferative effects both in vitro and in vivo, especially on HepG2 cells. A combination of drugs with additive and synergistic effects demonstrated a moderate-to-good cytotoxic activity against human cancer cells and obviously enhanced selectivity towards HepG2 compared to DDP and DOX, which suggested that Gardenblue anthocyanin extract may be developed as a potential antiproliferative agent by combining it with drugs in the future. The mechanism for this may be that Gardenblue anthocyanins interact with DNA in an intercalation mode, which could change or destroy DNA, cause apoptosis and inhibiting cancer cell proliferation. These findings require further research of more possible mechanisms. However, the antiproliferative effect decreases according to when the bioavailability of anthocyanins is low, thus further work on bioavailability will be needed in future. This research may contribute to the future development of antiproliferative drugs from natural foods or plants.

## Data Availability

Data are available upon request.

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
