# Peer review of "The Extraction and High Antiproliferative Effect of Anthocyanin from Gardenblue Blueberry"

_molecules, 2023, doi:10.3390/molecules28062850_

Round 1

Reviewer 1 Report

It is an interesting and important work, evidencing the capacity against cancer cell viability and its antitumor capacity in the in vivo model. Moreover, it shows that the extract does not act equally on all the cancer cell lines used, but that HepG2 cells are the most sensitive. However, the manuscript has some room for improvement:

1. to include the selectivity index of the compounds evaluated on cancer cells with respect to HUVEC non-malignant cells; 2. to include the selectivity index of the compounds evaluated on cancer cells with respect to HUVEC non-malignant cells.

2. In the introduction, considering that the study aims to explore the ability to inhibit cancer cell proliferation, the introduction lacks background or state of the art on the anticancer properties of blueberry and/or its main components. It is mainly based on other biological activities beneficial to health but not in the context of cancer. In addition, mention with respect to the drugs used why they were selected and why it is important to make the combination with the natural product obtained from blueberry.

3. In the methodology, mention the name of the reference standards used to identify the compounds of interest.

4. The MTT assay allows the evaluation of cell viability, it is not an assay to determine antitumor activity because the cultures are of neoplastic cells in monolayer.  

5. In the apoptosis assay, report how many events were analyzed and by which software and type of cytometer they were analyzed.

6. Why the in vivo experiment did not include a group treated with DPP or DOX alone and combined with Gardenblue anthocyanins?

7. Correct, the molecules analyzed and reported in Tables 1 and 2 do not refer to nutrients, but to phytochemicals such as total phenols and polyphenols like anthocyanins, ellagic acid and total flavonoids. Nutrients correspond to carbohydrates, lipids, proteins, vitamins and minerals, which were not analyzed in this research.

8. The MTT assay performed does not report the antiproliferative effect, but rather the ability to inhibit cell viability based on the IC50. 

9. The discussion lacks why the Gardenblue berry presented greater antiproliferative activity with respect to the other blueberries if they are all of the same genus Vaccinium and were collected in the same period and place.

 A comparative analysis of selectivity and sensitivity of the cell lines used with the anthocyanin extract alone and in combination should also be included, resulting effective only in HepG2 cells. The mechanism of action of the drugs used and how the effect of the extract in combination with doxorubicin and cisplatin is enhanced are also not compared. The authors do not mention if this berry can then be indicated or recommended for this type of cancer and not for the other cancer models studied in this project.

From the point of view of mechanisms, the authors should establish the relationship between apoptosis, the increase of intracellular ROS and DNA damage.

10. In the conclusion, it should be improved because an analysis of nutritional composition was not performed, phytochemicals of the (poly)phenol type were quantified and identified.

Reviewer 2 Report

In this study, 65 varieties of blueberries were investigated for their phytochemical profiles and antiproliferative effects on several cancerous cell lines showing good antiproliferative effect against HepG2 cells in vitro and in vivoThe paper provides an interesting analysis, however, several suggestions need to be addressed:

- English should be revised throughout

- some sentences require references

Abstract should be rephrased - the extract did not have a “strong antiproliferative effect” on all cancerous cells

Acronyms/Abbreviations should be defined the first time they appear in the abstractthe main textthe first figure or table 

Line 50-61: the authors could mention the type of experiments, e.g., in vitro, in vivo...

Lines 239-241: why do you say Gardenblue has a higher anthocyanin content than Lanmei 1”? when later is stated that “Lanmei 1 blueberries have an anthocyanin content of 443.08 mg/g and Gardenblue only 2.59 mg/g”! It is not mentioned if in Lanmei 1 is in dried powder.

Line 268: “rhamnoside

Lines 281-294: the paragraph should be rephrased; please do not repeat table data but emphasize or summarize the most important observations

Lines 295-300: same in this paragraph; “which revealed that a combination of drugs can improve the treatment effect”?! that happened only against HepG2 cells

Line 310: “Figure 2”

Lines 313-314: no need for “respectively” here

Line 342: “L1 (25 μg/mL) incubated with HepG2 cells”!? please correct 

Lines 354-355: the sentence should be rephrased

Line 416: “Yang et al.”

Discussion: this whole section is weak; it should be restructured, every assay should be discussed and compared to similar ones on the same plant matrix or related ones. For example in Line 333 you say “Linduced ROS production”: is that good or bad in cancerous cells? Please explain.

The in vivo results should be discussed and correlated with previously published studies; there is no mention of in vivo results in this section

In Lines 427-429: it should be mentioned that besides anthocyanins, ellagitannins and their gut microbiota-derived metabolites, other important bioactive molecules found in blueberries, were shown to trigger autophagy in the human colorectal cancer cells and to induce apoptosis by increasing the expression of proapoptotic proteins p21 and p53 and decreasing the anti-apoptotic protein expression of Bcl-2 (doi: 10.3390/foods12020270). Moreover, the proapoptotic effect was through downregulation of the PI3K/AKT signaling pathway.

Conclusions: this section is too general; authors should extend this sectionsummarize the results, and mention the limitations of their study, as well as the scope for future research.

Round 2

Reviewer 2 Report

The authors addressed the questions and suggestions and the manuscript has been improved.

Two minor suggestions:

- all assays and methods need citation if not yours 

- please check the tables; there are two "Table 2."

Author Response

  1. all assays and methods need citation if not yours 

Reply: Most of assays and methods are referred and improved from our previous work. We have added the citation, thank you.

  1. please check the tables; there are two "Table 2."

Reply: Sorry, we have revised all tables, please see the manuscript, thank you.